# Glial A_2B_ Adenosine Receptors Modulate Abnormal Tachykininergic Responses and Prevent Enteric Inflammation Associated with High Fat Diet-Induced Obesity

**DOI:** 10.3390/cells9051245

**Published:** 2020-05-18

**Authors:** Vanessa D’Antongiovanni, Laura Benvenuti, Matteo Fornai, Carolina Pellegrini, Renè van den Wijngaard, Silvia Cerantola, Maria Cecilia Giron, Valentina Caputi, Rocchina Colucci, Gyorgy Haskó, Zoltán H. Németh, Corrado Blandizzi, Luca Antonioli

**Affiliations:** 1Department of Clinical and Experimental Medicine, University of Pisa, 56126 Pisa, Italy; v.dantongiovanni@gmail.com (V.D.); laura.benvenuti962@gmail.com (L.B.); mfornai74@gmail.com (M.F.); lucaant@gmail.com (L.A.); 2Department of Pharmacy, University of Pisa, 56126 Pisa, Italy; carolina.pellegrini87@gmail.com; 3Tytgat Institute for Liver and Intestinal Research, Department of Gastroenterology and Hepatology, Academic Medical Center, 1105 Amsterdam, The Netherlands; r.vandenwijngaard@amc.uva.nl; 4Department of Pharmaceutical and Pharmacological Sciences, University of Padova, 35131 Padova, Italy; silvia.cerantola.2@gmail.com (S.C.); ceci.giron@gmail.com (M.C.G.); rocchina.colucci@unipd.it (R.C.); 5APC Microbiome Ireland, University College Cork, T12 YN60 Cork, Ireland; valekap@gmail.com; 6Department of Anesthesiology, Columbia University, New York, NY 10032, USA; nemeth1z@gmail.com; 7Department of Surgery, Morristown Medical Center, Morristown, NJ 07960, USA

**Keywords:** adenosine A_2B_ receptors, colonic motor dysfunction, enteric glia, enteric inflammation, glial cell derived neurotrophic factor, interleukin-1β, obesity, substance P, tachykininergic contraction, toll-like receptor 4

## Abstract

The role played by adenosine A_2B_ receptors (A_2B_Rs) in the regulation of enteric glial cell (EGC) functions remains unclear. This study was aimed at investigating the involvement of A_2B_Rs in the control of EGC functions in a model of obesity. C57BL/6 mice were fed with standard diet (SD) or high fat diet (HFD) for eight weeks. Colonic tachykininergic contractions were recorded in the presence of BAY60-6583 (A_2B_Rs agonist), MRS1754 (A_2B_Rs antagonist), and the gliotoxin fluorocitrate. Immunofluorescence distribution of HuC/D, S100β, and A_2B_Rs was assessed in whole mount preparations of colonic myenteric plexus. To mimic HFD, EGCs were incubated in vitro with palmitate (PA) and lipopolysaccharide (LPS), in the absence or in the presence of A_2B_R ligands. Toll-like receptor 4 (TLR4) expression was assessed by Western blot analysis. Interleukin-1β (IL-1β), substance P (SP), and glial cell derived neurotrophic factor (GDNF) release were determined by enzyme-linked immunosorbent assay (ELISA) assays. MRS1754 enhanced electrically evoked tachykininergic contractions of colonic preparations from HFD mice. BAY60-6583 decreased the evoked tachykininergic contractions, with higher efficacy in HFD mice. Such effects were blunted upon incubation with fluorocitrate. In in vitro experiments on EGCs, PA and LPS increased TLR4 expression as well as IL-1β, GDNF, and SP release. Incubation with BAY60-6583 reduced TLR4 expression as well as IL-1β, GDNF, and SP release. Such effects were blunted by MRS1754. The present results suggest that A_2B_Rs, expressed on EGCs, participate in the modulation of enteric inflammation and altered tachykininergic responses associated with obesity, thus representing a potential therapeutic target.

## 1. Introduction

Enteric glial cells (EGCs), the major component of the enteric nervous system (ENS), play a pivotal role in the regulation of intestinal homeostasis [1,2]. Indeed, such cells are distributed across all layers of the intestinal wall, where they control motility, barrier function, and enteric inflammation [1,2]. On the basis of this knowledge, the enteric glia is emerging as a new frontier in neuro-gastroenterology, representing a potential therapeutic target in several gastrointestinal (GI) disorders [3]. In this regard, several studies have shown that morphofunctional alterations of EGCs participate in the onset of intestinal neuronal alterations and maintenance of inflammatory conditions [2,4], suggesting their putative involvement in gut abnormalities under adverse conditions, including obesity.

Obesity is a pathological condition characterized by abnormal fat accumulation, as a result of a disequilibrium between energy intake and its consumption [5]. Of interest, such a pathological condition is characterized by low-grade systemic inflammation, which seems to be a common root to the onset and progression of several comorbidities, such as hypertension, cardiovascular disease, type 2 diabetes, fatty liver disease, and cancer [5]. Several studies have reported that adipocytes and adipose tissue-associated macrophages are important sources of pro-inflammatory mediators, such as interleukin (IL)-1, IL-6, IL-8, IL-18, tumor necrosis factor (TNF), and monocyte chemoattractant protein-1 (MCP)-1, thus contributing to insulin resistance and chronic obesity-induced metabolic dysfunctions [6]. There is also increasing clinical evidence that obesity is closely related with chronic GI disturbances, including gastroesophageal reflux, irritable bowel syndrome, diarrhea, and constipation [7,8]. The mechanisms underlying the onset and development of such disorders in the setting of obesity are still scarcely investigated. However, a recent paper from our group showed that EGCs are involved in the development of enteric motor disorders associated with a high fat diet (HFD), through an increase in the release of pro-inflammatory mediators and tachykininergic neurotransmission [9].

The endogenous mediators through which the enteric glia contributes to functional bowel disturbances remain presently unclear. There is, however, evidence showing that EGCs are a source of purines (adenosine triphosphate, ATP; adenosine diphosphate, ADP, and adenosine) and express a wide range of purinergic receptors, including adenosine A_2B_ receptors (A_2B_Rs) [10,11,12,13,14,15]. In particular, A_2B_R represents the adenosine receptor subtype mainly expressed both in human and rodent enteric glia when compared with A_1_, A_2A_, and A_3_ receptor [12,16], and appears to exert a pivotal role in the maintenance of homeostatic conditions in the digestive tract [17]. Recent studies have paid attention to the role of A_2B_Rs in the pathogenesis of altered colonic neuromuscular functions associated with obesity [18,19,20,21]. In particular, Antonioli et al. showed that A_2B_Rs, modulating the activity of excitatory tachykininergic nerves, were involved in colonic dysmotility associated with obesity [18,21]. However, the involvement of A_2B_Rs in shaping EGC functions in the presence of obesity remains unclear. On the basis of this background, the present study aimed at investigating the role played by A_2B_Rs in the modulation of EGC functions in a murine model of diet-induced obesity, in order to identify novel pharmacological therapeutic targets potentially suitable for the management of GI motor dysfunctions associated with obesity.

## 2. Materials and Methods

### 2.1. Experiments on Animals

#### 2.1.1. Animals and Diet

Male C57BL-6J mice, six weeks old, were employed throughout the study. The animals were fed with standard laboratory chow and tap water ad libitum, and were not employed for at least one week after their delivery to the laboratory. They were housed, three in a cage, in temperature-controlled rooms on a 12 h light cycle at 22–24 °C and 50–60% humidity. The experiments were approved by the Ethical Committee for Animal Experimentation of the University of Pisa and the Italian Ministry of Health, and were in full compliance with the European guidelines for the handling and use of experimental animals. The animals were purchased from ENVIGO S.r.l. (San Pietro al Natisone UD, Italy). To induce obesity, one group of mice was fed with HFD for eight weeks. The HFD (Altromin International, Germany; C 1090-60) provided 5.1 kcal/g, with 60.8% kcal as fats, 18.3% kcal as proteins, and 21.4% kcal as carbohydrates. The standard diet (Altromin International, Germany; SD, C 1090-10) provided 3.1 kcal/g, with 18% kcal as fats, 24% kcal as proteins, and 58% kcal as carbohydrates. Body weight and food intake were assessed once a week. Food intake was calculated by weighing all remnants of pellets at 11:00 and subtracting this value from the initial weight. At the end of the study, the animals were anaesthetized and euthanized.

#### 2.1.2. Recording of Colonic Contractile Activity

The contractile activity of colonic longitudinal smooth muscle preparations was recorded as previously described by Antonioli et al. [21]. Briefly, after euthanization, the distal colon was collected and immediately placed into a Petri dish containing warm pre-oxygenated Krebs solution. Colonic specimens were then cut along the longitudinal axis into strips of almost 4 mm width and 10 mm length. The preparations were set up in organ baths containing Krebs solution at 37 °C, bubbled with 95% O_2_ + 5% CO_2_, and connected to isometric transducers (resting load = 0.5 g). The Krebs solution had the following composition (mM): NaCl 113, NaHCO_3_ 25, glucose 11.5, KCl 4.7, CaCl_2_ 2.5, KH_2_PO_4_ 1.2, MgSO_4_ 1.2 (pH 7.4 ± 0.1). A pair of coaxial platinum electrodes was positioned at a distance of 10 mm from the longitudinal axis of each preparation to deliver electrical stimulations by a BM-ST6 stimulator (Biomedica Mangoni, Pisa, Italy). Each colonic preparation was allowed to equilibrate for at least 30 min, with intervening washes at 10 min intervals. At the end of the equilibration period, each preparation was repeatedly challenged with electrical stimuli, and the experiments started when reproducible motor responses were obtained (usually after two or three stimulations). Mechanical activity was recorded by BIOPAC MP150 (2Biological Instruments).

#### 2.1.3. Design of Functional Experiments

The first set of experiments was performed to examine the role of enteric glia in colonic tachykininergic NK_1_-mediated contractions. Contractions were recorded from colonic preparations maintained in Krebs solution containing 100 μM N^ω^-nitro-L-arginine methylester (L-NAME), 10 μM guanethidine, 1 μM atropine, 1 μM GR159897 (NK_2_ receptor antagonist), and 1 μM SB218795 (NK_3_ receptor antagonist), either in the absence or presence of the selective gliotoxin fluorocitrate (FC, 50 µΜ, Sigma-Aldrich F9634). FC is widely used to inhibit the tricarboxylic acid enzyme aconitase, determining the pharmacological blockade of glial cells through an alteration of their energy-dependent processes [22,23].

The second and third series aimed at investigating the effects of BAY60-6583 (A_2B_R agonist, Sigma-Aldrich SML1958) and MRS1754 (A_2B_R antagonist, Tocris 2752) in the absence or presence of FC. Electrically evoked contractions were recorded from colonic preparations maintained in Krebs solution containing 100 μM L-NAME, 10 μM guanethidine, 1 μM atropine, 1 μM GR159897, 1 μM SB218795, and 1 μM BAY60-6583 or 10 nM MRS1754, either in the absence or presence of 50 µΜ FC. Concentrations were selected in accordance with previous studies [9,21].

#### 2.1.4. Immunohistochemistry on Colonic Whole Mount Preparations

Fresh isolated distal colonic 3 cm segments from SD and HFD mice were gently flushed with warm (37 °C) Krebs solution to remove any luminal content, and incubated with or without 50 µΜ FC for 1 h in organ baths containing Krebs solution at 37 °C, bubbled with 95% O_2_ + 5% CO_2_ [9,24]. The colonic segments were then fixed with 4% paraformaldehyde in phosphate buffered saline (PBS) for 2 h at room temperature. After 3 × 15 min washes in PBS, the colonic segments were cut in 0.5 cm pieces, and opened along the mesenteric border to obtain preparations consisting of the longitudinal smooth muscle with attached myenteric plexus (LMMP), as previously described [24]. The colonic LMMP preparations were gently stretched and pinned down on a wax support, permeabilized in PBS with 0.3% Triton X-100 (PBT), and blocked with 2% bovine serum albumin (BSA) dissolved in PBT for 1 h at room temperature [24,25]. The LMMP preparations were then incubated overnight at room temperature with guinea pig polyclonal anti-rat S100β (1:100; Synaptic Systems, Göttingen, Germany), rabbit polyclonal anti-human A_2B_R (1:50; Merck Life Science, Milan, Italy), and mouse biotin-conjugated anti-human-HuC/D (1:100; Thermo Fisher Scientific, Milan, Italy) primary antibodies. The LMMP preparations were then washed with PBT and incubated with goat anti-guinea pig IgG Alexa Fluor 488-conjugated (1:1000; Thermo Fisher Scientific Milan, Italy), streptavidin Alexa Fluor 555-conjugated (1:1000, Thermo Fisher Scientific, Milan, Italy), and DyLight 649-conjugated goat anti-rabbit IgG (1:500, Jackson ImmunoResearch Laboratories, Milan, Italy) for 1 h at room temperature. After three washes with PBT, the preparations were mounted on glass slides using a mounting solution (Citifluor AF1, SIC s.rl., Italia) and stored at −20 °C in the dark until analysis. Negative controls were obtained by incubation with isotype-matched control antibodies at the same concentration as the primary antibody and/or by pre-incubating each antibody with the corresponding control peptide (final concentration as indicated by manufacturer’s instructions). Images were acquired using a Zeiss LSM 800 confocal imaging system (Oberkoken, Germany) equipped with an oil-immersion 63× objective (NA 1.4). Z-series images (15 planes for each LMMP whole mount preparations) of 1024 pixels × 1024 pixels were processed as maximum intensity projections. All microscope settings were fixed to collect images below saturation and were constant for all the images. Fluorescence intensity (density index) of S100β and A_2B_R was assessed for each antigen by capturing 20 images per mouse. The intensity of staining for each antibody was expressed as the density index of labelling per myenteric ganglion area, as previously described [24], and was reported as mean ± SEM.

### 2.2. Experiments on Cultured Enteric Glial Cells

#### 2.2.1. Cell Culture

Rat-transformed EGCs were obtained from ATCC^®^ (EGC/PK060399egfr; ATCC® CRL-2690™; Manassas, VA, USA). The cells were grown and maintained in Dulbecco’s modified Eagle’s medium (DMEM) supplemented with 10% fetal bovine serum (FBS), 2 mM glutamine, and 100 unit/mL penicillin-streptomycin at 37 °C in a humidified atmosphere of 5% CO_2_.

#### 2.2.2. Palmitate-Bovine Serum Albumin Complex Preparation

A stock solution of palmitate (PA, Sigma-Aldrich) was prepared as previously described by Antonioli et al. [9]. Briefly, to obtain 200 mM of stock solution, PA was dissolved in pre-heated water and the mixture was incubated at 70 °C overnight with constant vortexing. The PA-BSA complex was prepared by mixing the stock solution to 10% BSA (Sigma-Aldrich). The formed complex was diluted in culture medium to prepare 400 μM PA treatment. The same final concentration of BSA (0.5% v/v) was maintained in all treated cells to avoid a differential protein binding effect on compounds.

#### 2.2.3. Stimulation Protocol

EGCs were seeded at a density of 1 × 10^6^ cells in a 100 mm Petri dish containing culture medium. To mimic the in vivo features of HFD exposure, cells were treated with 400 μM PA for four days and 10 μg/mL lipopolysaccharide (LPS, Sigma-Aldrich) for 24 h before the end of the experiment, in the absence or presence of 0.05 µM BAY60-6583 and/or 0.1 µM MRS1754. To minimize interferences by extracellular adenosine, cells were incubated with 3 U/mL of adenosine deaminase (ADA) for 3 h before treatment with A_2B_ ligands. Controls were run in parallel. Concentrations were selected in accordance with previous reports [26,27,28,29,30].

For the experiments with the inhibitor of glial cell derived neurotrophic factor (GDNF) receptor signaling, cells were treated for 19 h with 100 µM RPI-1 (1,3-dihydro-5,6-dimethoxy-3-[(4-hydroxyphenyl)methylene]-H-indol-2-one; Merck/Calbiochem), a specific compound known to block the downstream signaling of GDNF receptor [31,32]. The details of all treatments are shown in Figure 1.

#### 2.2.4. Western Blot

Cells were lysed as described previously [33,34]. Proteins were quantified with the Bradford assay. Proteins were separated onto a pre-cast 4–20% polyacrylamide gel (Mini-PROTEAN^®^ TGX gel, Biorad) and transferred to polyvinylidene difluoride (PVDF) membranes (Trans-Blot^®^ TurboTM PVDF Transfer packs, Biorad). Membranes were blocked with 3% BSA diluted in Tris-buffered saline (TBS, 20 mM Tris-HCl, PH 7.5, 150 mM NaCl) with 0.1% Tween 20. Primary antibodies against β-actin (ab8227, Abcam) and toll-like receptor 4 (TLR4; ab22048, Abcam) were used. Secondary antibodies were obtained from Abcam (anti-mouse ab97040 and anti-rabbit ab6721). Protein bands were detected with enhanced chemiluminescence (ECL) reagents (Clarity^ᵀᴹ^ Western ECL Blotting Substrate, Biorad). Densitometry was performed by ImageJ software.

#### 2.2.5. Assessment of Interleukin-1β, Substance P, and Glial Cell Derived Neurotrophic Factor Production from EGCs

The release of interleukin (IL)-1β, substance P (SP), and GDNF into culture medium was measured by enzyme-linked immunosorbent assay (ELISA) kits (Abcam), according to the manufacturer’s protocols. After cell stimulation, the medium was collected and centrifuged for 5 min at 800 rpm to obtain cell-free supernatants. Supernatants (150 μl for IL-1β assay; 100 μl for SP and GDNF assays) were then used. Absorbance was normalized to the number of cells.

### 2.3. Statistical Analysis

Data are presented as mean ± SEM and analyzed by GraphPad Prism 7.0 (GraphPad Software Inc., San Diego, CA, USA). Statistical significances were determined by one-way analysis of variance (ANOVA) followed by Tukey’s post hoc test or Dunnett’s post hoc test. A *p*-value ≤ 0.05 was considered significantly different.

## 3. Results

### 3.1. Body Weight and Food Intake

A significant increase in body weight was observed in HFD, starting from week 4, as compared with SD mice (Figure 2A). No significant alterations in food intake were observed in both groups (Figure 2B).

### 3.2. A_2B_ Receptors on Enteric Glia Modulate Tachykininergic Contractions in HFD Mice

All the experiments were set up in Krebs solution containing 10 μM guanethidine, 100 μM L-NAME, 1 μM atropine, 1 μM GR159897, and 1 μM SB218795 in order to record electrically evoked NK_1_-mediated tachykininergic contractions. Under these conditions, colonic preparations from HFD mice displayed a significant increment of electrically evoked tachykininergic contractions, when compared with SD animals, which was blunted by the presence of FC in the Krebs solution (Figure 3A,B).

In colonic preparations from HFD mice, incubation with the A2BR agonist BAY60-6583 (1 μM), either in the absence or presence of FC, significantly reduced the electrically evoked NK_1_-mediated tachykininergic contractions (Figure 3A). The incubation of colonic preparations from HFD mice with the A2BR antagonist MRS1754 (10 nM) induced a significant increment of NK_1_-mediated contractions (Figure 3B), which were blunted by co-incubation with FC (Figure 3B).

In colonic preparations from SD mice, the incubation with A_2B_R ligands or FC did not alter electrically evoked NK_1_-mediated tachykininergic contractions (*p* = 0.98 with FC; *p* = 0.97 with BAY60-6583; and *p* = 0.99 with MRS1754, MRS1754 + FC, and BAY60-6583 + FC) (Figure 3A,B).

### 3.3. Effect of Fluorocitrate on A_2B_ Receptor and Glial Protein S100β Distribution

A_2B_R immunoreactivity was detected in colonic LMMPs from SD and HFD mice (Figure 4). In SD mice, A_2B_R immunofluorescence was found primarily in S100β^+^ myenteric glial cells (Figure 4A). Incubation with FC induced a significant increase in A_2B_R immunoreactivity (+21%; Figure 4B), mainly in neurons, while no appreciable changes were found in glial cells (Figure 4C).

In colonic LMMPs from HFD mice, we observed an increase in glial reactivity characterized by a marked increase in S100β immunofluorescence along with an increased A_2B_R staining (+38% and +37%, respectively; Figure 4B,C). Ex vivo treatment with FC counteracted the glial A_2B_R changes observed in HFD tissues, as compared with SD colonic preparations (Figure 4A,B).

### 3.4. A_2B_ Receptors Modulate the Release of Substance P and Glial Cell Derived Neurotrophic Factor from Cultured EGCs

Incubation with PA plus LPS induced a significant increase in SP release, as compared with control cells, which was counteracted significantly by BAY60-6583. The effect of BAY60-6583 was antagonized significantly by MRS1754 (Figure 5A).

Treatment with PA and LPS determined a significant increase in the release of GDNF, as compared with control cells (Figure 5B). In the presence of PA and LPS, treatment with BAY60-6583 significantly reduced extracellular GDNF levels, as compared with control cells, whereas GDNF release was enhanced significantly by incubation with MRS1754 (Figure 5B).

Incubation with RPI-1, an inhibitor of GDNF receptor signaling, in the presence of PA and LPS, did not counteract the release of SP from EGCs, although it significantly reduced extracellular GDNF release (Figure 5C,D).

### 3.5. A_2B_ Receptors Modulate TLR4 Expression in Cultured EGCs Incubated with Palmitate and Lipopolysaccharide

Incubation of EGCs with PA and LPS induced a significant increase in TLR4 expression (Figure 6). Under these conditions, treatment with BAY60-6583 significantly decreased TLR4 expression. In addition, the effect of BAY60-6583 was antagonized by MRS1754 (Figure 6).

### 3.6. A_2B_ Receptors Modulate Interleukin-1β Release Induced by Palmitate and Lipopolysaccharide in Cultured EGCs

Incubation of EGCs with PA and LPS induced a significant increase in IL-1β release, as compared with control cells (Figure 7). Such an increase was counteracted by BAY60-6583. Moreover, incubation with MRS1754 reversed the inhibitory effect of BAY60-6583 (Figure 7).

## 4. Discussion

Increasing evidence points to an active involvement of EGCs in controlling several homeostatic gut functions, including mucosal sensation, secretion, motility, and immune responses, as well as a pivotal role in the pathophysiology of enteric motor dysfunctions associated with inflammatory conditions [9,35]. In line with this view, our previous study highlighted the contribution of enteric glia in the onset of colonic motor alterations associated with HFD-induced obesity, through an increase in tachykininergic activity and release of pro-inflammatory mediators (i.e., IL-1β) [9].

Several lines of evidence have identified EGCs as an “ideal hub” able to interact with the surrounding environment, receiving and processing inputs arising from neurons and other non-neuronal cells (i.e., epithelial, smooth muscle, and immune cells) through the release of a wide range of molecular factors, including adenosine [11,15,36]. Indeed, EGCs appear to be highly responsive to purines, through the expression of receptors for ATP and its derivatives, such as ADP and adenosine, which are useful to shape glial immuno-inflammatory actions, but are still scarcely investigated regarding their influence on gut physiology and pathophysiology [37]. In this context, based on the evidence describing a marked presence of the A_2B_R subtype on enteric glia [12,16], we designed the present study to elucidate the role played by colonic A_2B_Rs in the modulation of EGC functions in a murine model of diet-induced obesity.

Our experiments, performed on colonic tissues from HFD mice and cultured EGCs, pointed to three major novel findings: (1) a modulatory role played by endogenous adenosine, via glial A_2B_Rs, on colonic SP-mediated contractions; (2) a modulatory effect of A_2B_R ligands on SP release and a regulatory action on GDNF release, a neurotrophic factor involved in ENS development and neuronal activity as well as the survival and differentiation of enteric neurons; and (3) a critical role of A_2B_Rs in shaping the neuroimmune communications of enteric glia through the modulation of TLR4 expression and IL-1β release.

In the present study, mice fed with HFD for eight weeks showed a marked increase in body weight, followed by a marked alteration of several metabolic indexes, such as an increase in blood glucose, cholesterol, and triglycerides (data not shown), thus corroborating the suitability of this experimental model. As a first step, we attempted to determine the involvement of A_2B_Rs, expressed on enteric glia, in the colonic motor alterations associated with obesity. To pursue this goal, we performed a series of functional in vitro experiments, where colonic preparations were incubated with A_2B_R ligands in the absence or presence of the gliotoxin FC. The tachykininergic contractions of colonic preparations from HFD mice were significantly enhanced by the pharmacological blockade of A_2B_R. Of note, when incubated with FC, the effects of the A_2B_R antagonist no longer occurred, thus indicating an inhibitory action of endogenous adenosine on NK_1_-mediated contraction via glial A_2B_Rs. The incubation of colonic preparations with the selective A_2B_R agonist BAY60-6583 reduced electrically-evoked tachykininergic contractions, mainly in tissues from HFD mice. Interestingly, the inhibitory effect of BAY60-6583 alone was similar in magnitude to that observed upon incubation of colonic preparations with FC, thus suggesting the presence of inhibitory A_2B_Rs at the glial level.

Subsequently, we examined the distribution of A_2B_Rs and the glial protein S100β in colonic LMMP whole mount preparations. Our results in SD mice showed that A_2B_Rs are expressed mainly in glial cells, and that the FC-mediated inhibition of glial activity also determined an increased staining in neurons, suggesting a possible involvement of A_2B_Rs in the glial-neuronal crosstalk at the level of the ENS [21], as previously determined in the central nervous system (CNS), where they appear to be involved in the regulation of astrocyte-derived BDNF (brain-derived neurotropic factor) production [38]. An increase in EGC activity is usually defined as reactive gliosis, a process denoting a broad spectrum of potential responses of glial cells to ENS insults [9,24]. HFD affected both S100β and A_2B_R immunoreactivity, determining a marked increment of both antigens in the colonic ENS, suggesting a possible link between reactive gliosis and A_2B_R-mediated responses. However, in the presence of FC, the ENS dysfunction was corrected mostly by blunted EGC activation, suggesting that the HFD-induced increase in A_2B_R immunoreactivity depends mostly on glial adaptive changes. Taken together, functional data and immunohistochemical findings indicate that A_2B_Rs on enteric glia are markedly involved in the regulation of colonic contractions in obese mice, exerting a tonic inhibitory control on motor tachykininergic pathways.

The second part of the study aimed at characterizing the molecular mechanisms underlying the modulatory actions of A_2B_R on EGCs. For this purpose, cultured EGCs were incubated with PA (the major source of fat in our hypercaloric diet) and LPS (to reflect the increase in endotoxemia associated with HFD intake), in order to mimic the in vivo features of HFD exposure. Of interest, consistent with our functional studies, we observed that the incubation of EGCs with PA and LPS resulted in a marked increase in SP release, which was blunted by the pharmacological activation of A_2B_Rs. Despite an inhibitory action of adenosine on SP release being reported previously [39,40], our experiments provide evidence, for the first time, of a modulatory control exerted by A_2B_Rs on glial SP release. This is an interesting point, as it is well recognized that the enteric glia represents a relevant source of SP under inflammatory conditions, where it holds an active role in the pathogenesis of enteric dysmotility [9,41].

In parallel, we paid attention to the putative modulatory action of A_2B_Rs on GDNF, a glial-derived neurotrophic factor indicated by several studies as relevantly involved in the regulation of intestinal epithelial and neuronal functions under inflammatory conditions [42,43]. GDNF expression and release are markedly increased in the presence of several pathological conditions, including obesity, likely in an attempt at maintaining the integrity of intestinal epithelial barrier and mitigating the inflammatory response [43,44]. However, it is also noteworthy that a chronic increase in GDNF has been shown to exert detrimental effects in the gut, eliciting morphofunctional alterations of myenteric and submucosal neurons, with a consequent unsettlement of enteric motility and bowel transit [45]. Accordingly, our in vitro experiments, performed under conditions mimicking obesity, revealed a significant increase in GDNF release from EGCs, which was blunted by incubation with the A_2B_R agonist. As a previous study, performed on rat colonic neuromuscular preparations, reported a stimulant effect of GDNF on SP release [46], we performed a set of in vitro experiments aimed at verifying the occurrence of such interplay in obesity as well. To pursue this goal, we incubated EGCs with PA and LPS in the absence or presence of RPI-1, an inhibitor of GDNF receptor signaling (RET receptor tyrosine kinase), observing that RPI-1 failed to counteract the increased release of SP from EGCs, and thus ruling out a possible modulatory action of GDNF on SP release. Of note, this finding appears to be in contrast with the above mentioned study [46]. However, it is noteworthy that our experimental conditions were focused only on EGCs, thus not allowing to investigate all possible interactions among the various cell populations of the enteric neuromuscular compartment. Indeed, although, from our results, a direct link between GDNF and SP does not appear to occur at a glial level, it is conceivable that an indirect interplay might take place among glial GDNF and SP release at the level of myenteric neurons. Clearly, further experiments are needed to better investigate the existence of these putative mechanisms.

EGCs, beyond behaving as a structural and trophic support to neurons, appear to be a sort of bridge between immune-inflammatory cells and ENS [37]. Indeed, EGCs, through the acquisition of a pro-inflammatory phenotype, with the consequent production of cytokines (i.e., IL-1β, IL-6, INF-γ, TNF) and release of neurotrophic factors (i.e., BDNF, GDNF and nerve growth factor, NGF), contribute relevantly to the health and disease status of the digestive tract [37,41]. In this regard, our experiments highlighted a role of enteric glia in supporting the pro-inflammatory environment in the presence of obesity. In support of this view, we observed that, when exposed to PA and LPS, EGCs reacted with an increase in TLR4 expression and enhanced IL-1β release. Previous studies reported a stimulant effect of palmitic acid and other saturated fatty acids on TLR4 receptors expressed on human dendritic cells, leading to an increase in pro-inflammatory cytokine release (TNF and IL-1β) [47]. Likewise, a recent article from our group provided evidence of the effect of PA and LPS on the pro-inflammatory polarization of EGCs, characterized by an increase in IL-1β release [9]. This is an intriguing point, as IL-1β is emerging as a critical player in several obesity-related comorbidities (i.e., insulin resistance, meta-inflammatory condition, cardiovascular diseases) [48,49], and thus the enteric glia could participate in the maintenance of such disorders. We also observed that, when pharmacologically stimulated, A_2B_Rs counteracted the glial pro-inflammatory responses, reducing both TLR4 expression and IL-1β release from cultured EGCs. Despite that this finding represents the first evidence of an immunomodulatory action of A_2B_Rs on glial cells, it is in line with previous studies describing a reduction of TLR4 expression and a decrease in the release of pro-inflammatory cytokines (i.e., IL-1β, TNF, IL-6, and IL-12) from human and murine macrophages incubated with A_2A_ or A_2B_ receptor agonists [50,51,52,53,54,55]. Interestingly, a recent paper by Csóka et al. elegantly demonstrated that the stimulation of A_2B_Rs suppressed the deleterious inflammatory and metabolic activation of macrophages induced by free fatty acids [56]. In particular, the authors reported that the A_2B_R engagement counteracted the release of cytokines (i.e., IL-6 and TNF) and induced the polarization of such immune cells towards an anti-inflammatory phenotype [56]. On the basis of this knowledge, it is conceivable that the stimulation of glial A_2B_Rs, beyond reducing the IL-1β release, could be able to modulate the release of other pro-inflammatory cytokines, such as IL-6 and TNF, also under our experimental conditions. However, future studies are needed to better clarify the role of adenosine receptors in the modulation of the complex network of cytokines released from EGCs under inflammatory conditions.

Taken together, our results, obtained from colonic tissues isolated from HFD mice and cultured EGCs, point to the concept that enteric glial alterations, occurring in the presence of obesity, sustain both bowel motor dysfunctions and enteric inflammation, through GDNF, SP, and IL-1β release. In this context, we also suggest that a hyperactivity of endogenous adenosine, through the activation of glial A_2B_Rs, exerts an inhibitory action on GDNF, SP, and IL-1β release.

When considering the inflammatory role played by the enteric glia, it must be considered that SP, beyond acting as an important neurotransmitter involved in the regulation of GI functions (i.e., motility, secretion, and visceral sensitivity), also takes an important part in the immuno-inflammatory response, driving the release of several cytokines [35,57]. For this reason, the results of the present study allow to hypothesize that the pharmacological stimulation of glial A_2B_Rs can concomitantly ensure both the control of the release of mediators involved in the pathogenesis of colonic neuromuscular alterations (SP and GDNF) and the modulation of the production of pro-inflammatory mediators (IL-1β and SP) (Figure 8). In this regard, to better investigate the therapeutic potential of A_2B_Rs, we have designed additional experiments, based on the treatment of obese animals with selective A_2B_R ligands, with the purpose of shedding light on the putative beneficial effects of pharmacological A_2B_R modulation in managing both bowel motor dysfunctions and inflammation associated with obesity.

## Figures and Tables

**Figure 1 cells-09-01245-f001:**
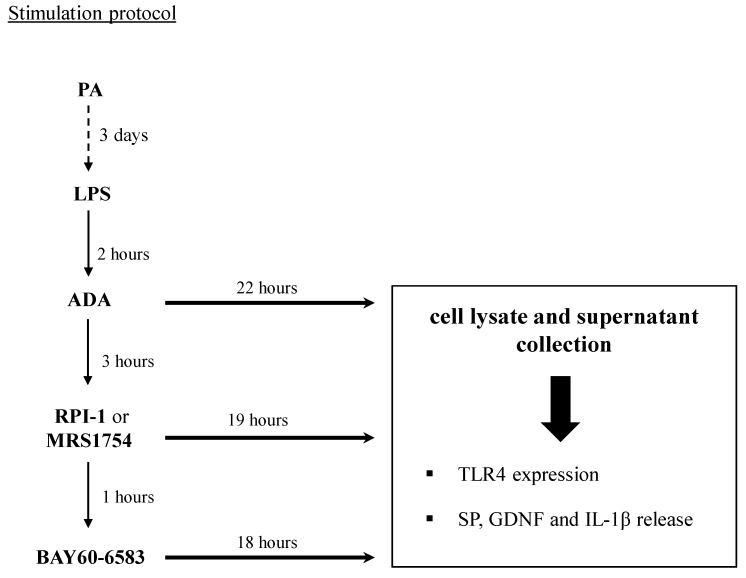
Schematic representation of the design of experiments on cultured enteric glial cells (EGCs). EGCs were treated for three days with palmitate (PA, 400 μM). Then, cells were incubated for 2 h with lipopolysaccharide (LPS, 10 μg/mL) before the addition of adenosine deaminase (ADA, 3 U/mL), to minimize interferences by extracellular adenosine. Under these conditions, cells were treated for 19 h with MRS1754 (0.1 µM, A_2B_R antagonist) or incubated for 1 h with MRS1754 before the addition of BAY60-6583 (0.05 µM, selective A_2B_R agonist, 18 h). For the experiments with the inhibitor of glial cell-derived neurotrophic factor (GDNF) receptor signaling, cells were treated with RPI-1 (100 µM, 1,3-dihydro-5,6-dimethoxy-3-[(4-hydroxyphenyl)methylene]-H-indol-2-one) for 19 h. On the fourth day, cells were lysed for analysis of toll-like receptor 4 (TLR4) expression and the culture media were collected for analysis of substance P (SP), GDNF, and interleukin (IL)-1β release.

**Figure 2 cells-09-01245-f002:**
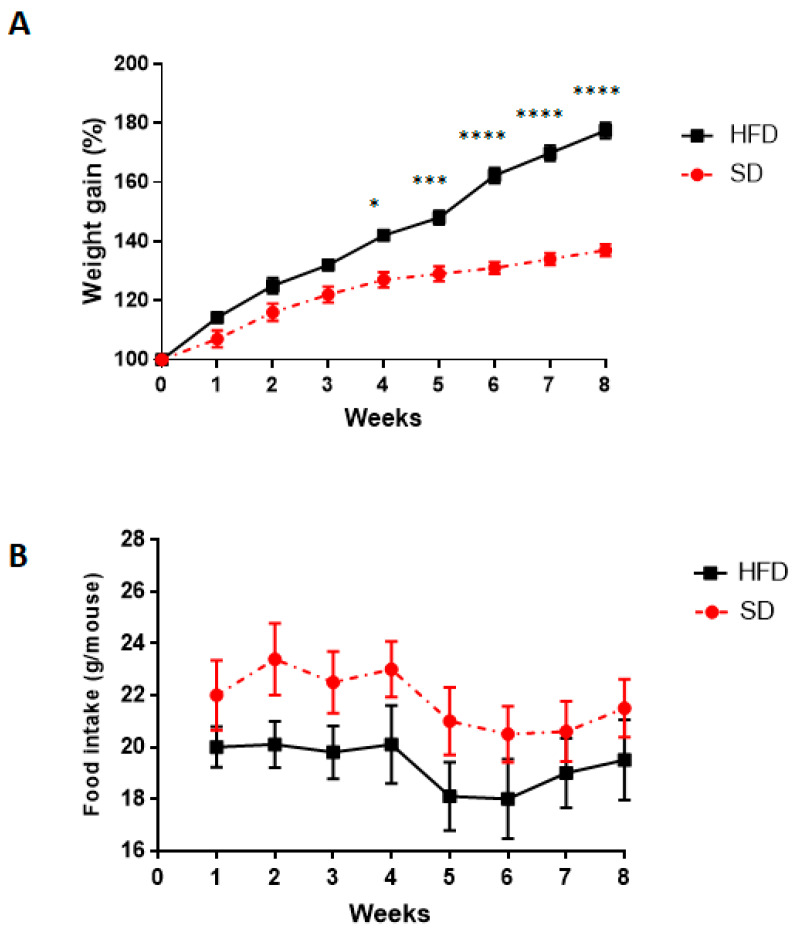
Effects of standard diet (SD) or high-fat diet (HFD) on (**A**) body weight gain and (**B**) food intake. Data are means ± SEM (n = 8). * *p* < 0.05, *** *p* <0.001, **** *p* <0.0001 versus SD.

**Figure 3 cells-09-01245-f003:**
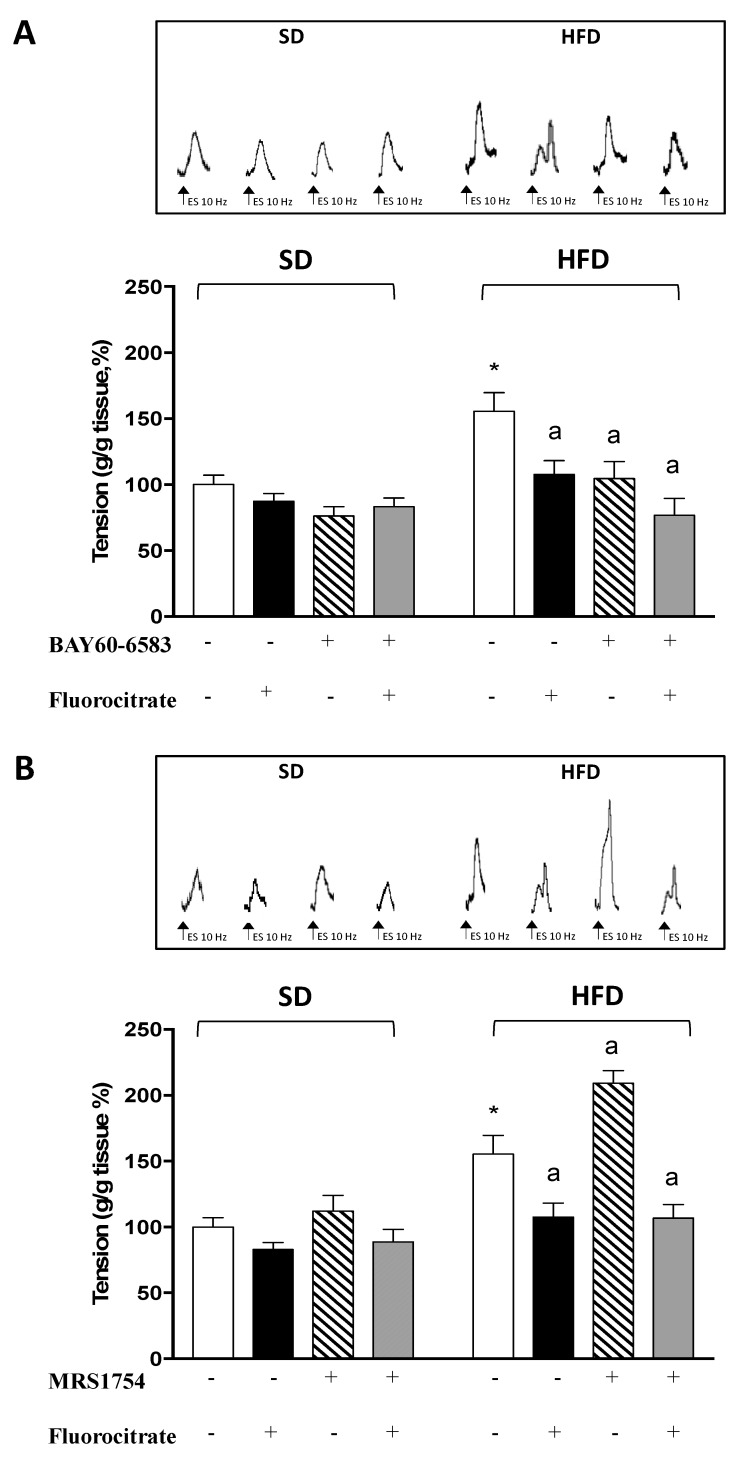
Effects of 1 μM BAY60-6583 (**A**) or 10 nM MRS1754 (**B**) on tachykininergic contractions elicited, in the absence or presence of fluorocitrate (50 μM, FC), by electrical stimulation of longitudinal smooth muscle preparations of distal colon from mice fed with standard diet (SD) or high fat diet (HFD). Colonic preparations were maintained in Krebs solution containing 100 μM N^ω^-nitro-L-arginine methylester (L-NAME), 10 μM guanethidine, 1 μM atropine, 1 μM GR159897 (NK_2_ receptor antagonist), and 1 μM SB218795 (NK_3_ receptor antagonist). Each column represents the mean ± SEM (n = 8). * *p* < 0.05 versus SD; ^a^
*p* < 0.05 versus HFD.

**Figure 4 cells-09-01245-f004:**
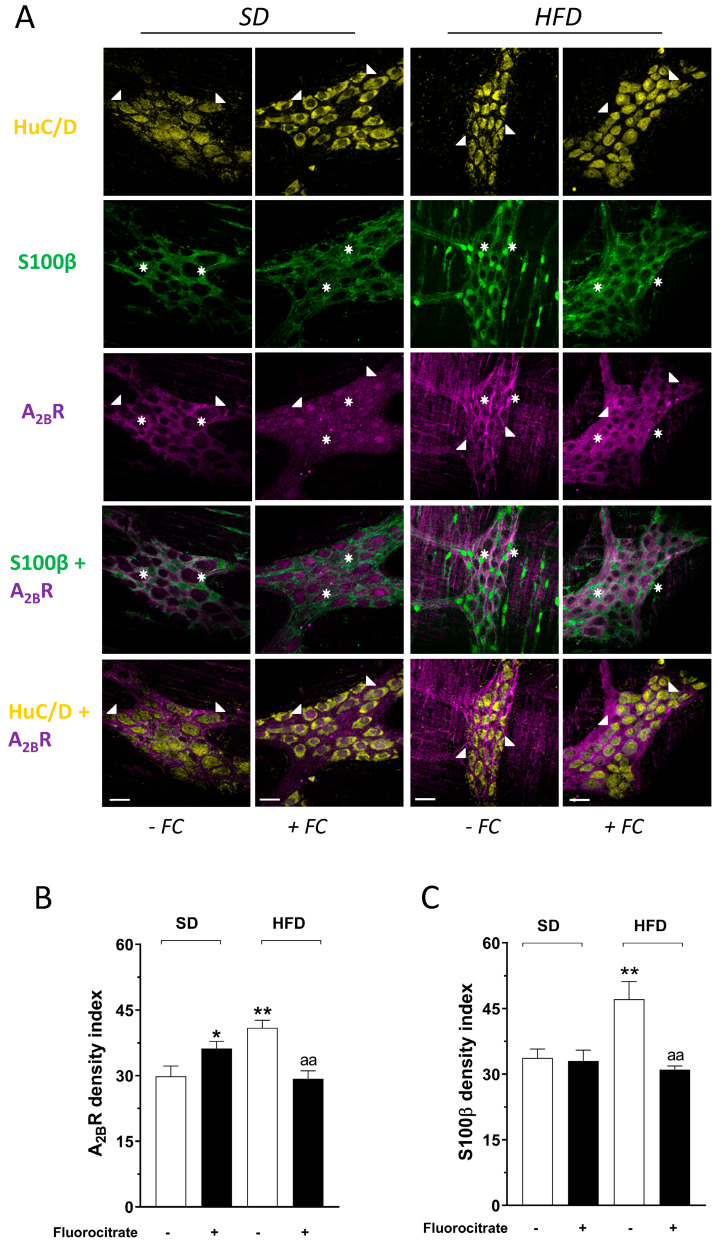
(**A**) Representative confocal microphotographs showing the distribution of the neural marker HuC/D (yellow), the glial protein S100β (green), and A_2B_Rs (magenta) in colonic longitudinal smooth muscle with attached myenteric plexus (LMMP) whole mount preparations from mice fed with standard diet (SD) or high-fat diet (HFD), in the absence or presence of 50 μM fluorocitrate (FC). White arrowheads indicate HuC/D^+^ and A_2B_R^+^ neurons, while white stars indicate S100β^+^ and A_2B_R^+^ glial cells. Scale bars = 22 μm; analysis of A_2B_R (**B**) and S100B (**C**) density index in colonic LMMP whole mount preparations from mice fed with SD or HFD, in the absence or presence of 50 μM FC. * *p* < 0.05, ** *p* < 0.01 versus SD without FC; ^aa^
*p* < 0.01 versus HFD; n = 5 mice per group.

**Figure 5 cells-09-01245-f005:**
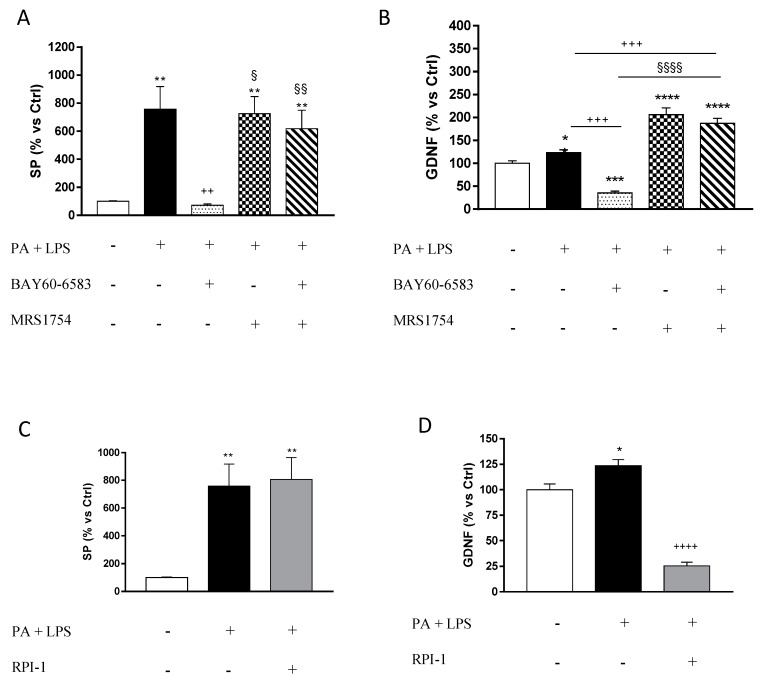
(**A**) Substance P (SP) release from cultured enteric glial cells (EGCs) treated with palmitate (PA, 400 μM) and lipopolysaccharide (LPS, 10 μg/mL), in the absence or presence of BAY60-6583 (0.05 µM) and/or MRS1754 (0.1 µM); (**B**) release of glial cell-derived neurotrophic factor (GDNF) from EGCs incubated with PA and LPS, in the absence or presence of BAY60-6583 and/or MRS1754; (**C**) SP release from EGCs treated with PA and LPS, in the absence or presence of RPI-1; (**D**) release of GDNF from EGCs incubated with PA and LPS, in the absence or presence of RPI-1. Data are means ± SEM. (n = 5). * *p* ≤ 0.05, ** *p* ≤ 0.01, *** *p* ≤ 0.001, **** *p* ≤ 0.0001 vs. Ctrl; ^+++^
*p* ≤ 0.001, ^++++^
*p* ≤ 0.001 vs. PA + LPS; ^§^
*p* ≤ 0.05, ^§§^
*p* ≤ 0.01, ^§§§§^
*p* ≤ 0.0001 vs. PA + LPS + BAY60-6583.

**Figure 6 cells-09-01245-f006:**
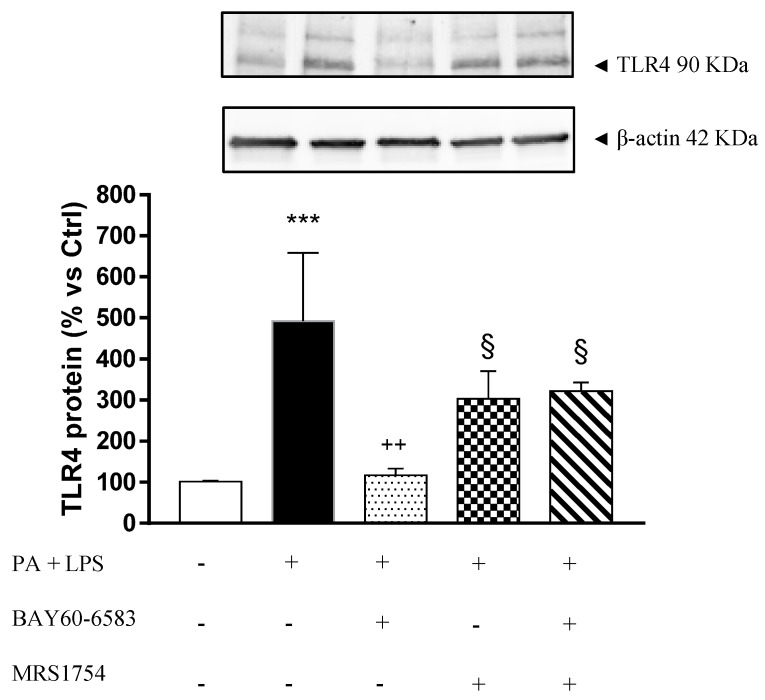
Representative blots and densitometric analysis of TLR4 expression assessed by Western blot assay in cultured enteric glial cells (EGCs) treated with palmitate (PA, 400 μM) and lipopolysaccharide (LPS, 10 μg/mL), in the absence or presence of BAY60-6583 (0.05 µM) and/or MRS1754 (0.1 µM). Data are means ± SEM (n = 5). *** *p* ≤ 0.001 vs. Ctrl; ^++^
*p* ≤ 0.01 vs. PA + LPS; ^§^
*p* ≤ 0.05 vs. PA + LPS + BAY60-6583.

**Figure 7 cells-09-01245-f007:**
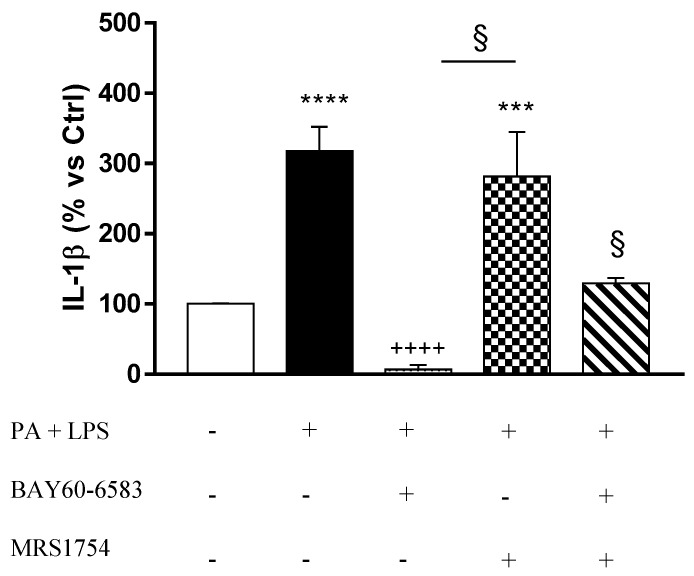
Interleukin (IL)-1β release from cultured enteric glial cells (EGCs) incubated with palmitate (PA, 400 μM) and lipopolysaccharide (LPS, 10 μg/mL), in the absence or presence of BAY60-6583 (0.05 µM) and/or MRS1754 (0.1 µM). Data are means ± SEM (n = 5). *** *p* ≤ 0.001, **** *p* ≤ 0.0001 vs. Ctrl; ++++ *p* ≤ 0.0001 vs. PA + LPS; § *p* ≤ 0.05 vs. PA + LPS + BAY60-6583.

**Figure 8 cells-09-01245-f008:**
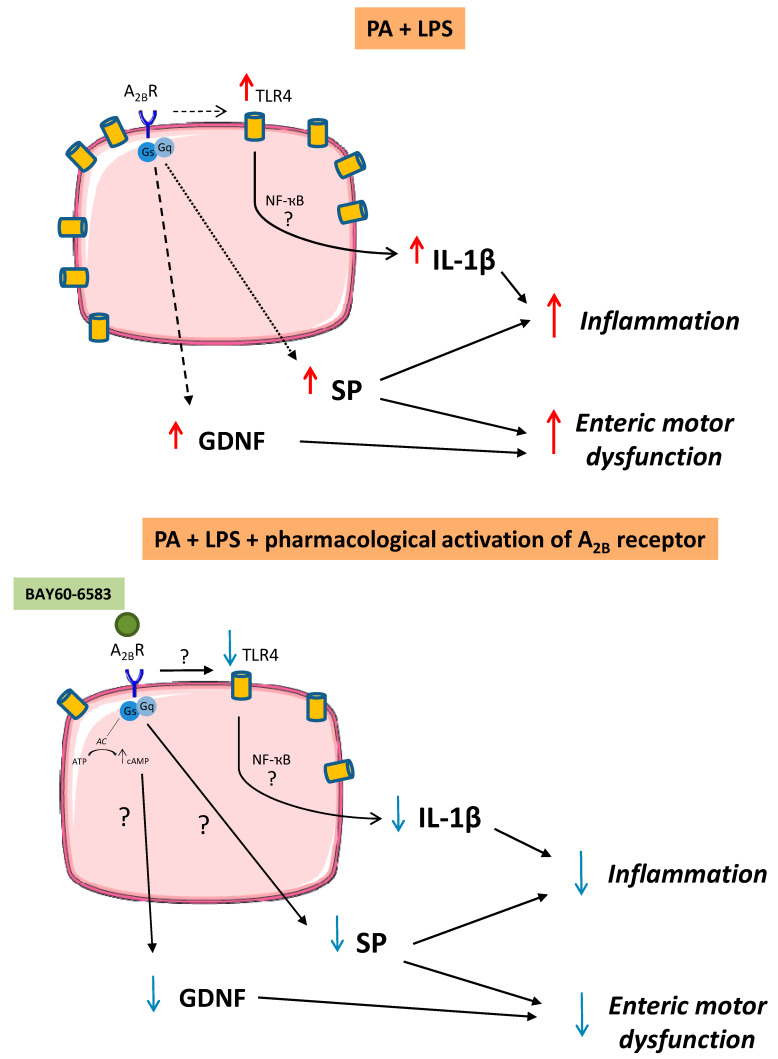
Schematic representation of the proposed role of A_2B_Rs in the modulation of enteric glial cell (EGC) activity under experimental conditions mimicking obesity. In vitro incubation of EGCs with palmitate (PA, 400 μM) and lipopolysaccharide (LPS, 10 μg/mL) elicited an increase in TLR4 expression as well as substance P (SP), glial cell-derived neurotrophic factor (GDNF), and interleukin (IL)-1β release, suggesting an involvement of these cells in supporting the enteric inflammation and abnormal tachykininergic enteric motor responses associated with obesity. The pharmacological stimulation of A_2B_Rs on EGCs, besides reducing TLR4 expression and IL-1β levels, can also counteract the abnormal increase in GDNF and SP release. A_2B_R: adenosine A_2B_ receptor; AC: adenylyl cyclase; ATP: adenosine triphosphate; cAMP: cyclic adenosine monophosphate; NF-ҡB: nuclear factor kappa-light-chain-enhancer of activated B cells.

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
