# Peer review of "Glial A2B Adenosine Receptors Modulate Abnormal Tachykininergic Responses and Prevent Enteric Inflammation Associated with High Fat Diet-Induced Obesity"

_cells, 2020, doi:10.3390/cells9051245_

Round 1

Reviewer 1 Report

The manuscript entitled “Glial A2B adenosine receptors modulate abnormal tachykininergic responses and prevent enteric inflammation associated with high fat diet-induced obesity” by D’Antongiovanni and co-workers focuses on the role played by colonic A2BRs in the modulation of enteric glial cell functions. For this purpose they used a murine model of diet-induced obesity and they performed in vitro experiments mimicking obesity (rat transformed enteric glial cell treated with palmitate and LPS). The manuscript is well written, it is easy to comprehend and well organized.

However, I have some comments:

-The authors used high fat diet model (8 weeks) to induce obesity in C57BL6/J mice. They should report body weight changes and food intake measure.

-The authors could add representative traces of contractile responses in Fig.2.

- A quantification of fluorescent intensity is required about the effect of fluorocitrate on A2B receptor and glial protein S100β distribution assessed by confocal microscopy.

- The authors show that A2B receptor agonist, BAY60-6583, reduces TLR4 expression as well as IL-1β, GDNF and SP release induced by palmitate and lipopolysaccharide in cultured rat-transformed enteric glial cells. Exposure to LPS induces the expression of several inflammatory cytokines in enteric glia including IL-6 and TNF-α. Does the A2B agonist, BAY60-6583, modulate the levels of these cytokines in the in vitro model used?

- The major signalling pathway of A2BAR is suggested to involve adenylyl cyclase. Could the authors explain the intracellular signalling pathways coupled to the A2B receptor activation in the effects observed? Does the selective targeting of TLR4 on enteric glial cells cause a downstream inhibition of nuclear factor kappa B-dependent inflammation?

-The authors could add the concentration used for BAY60-6583 and MRS1754 in all figures or in their legend to help the reader as well as for PA, LPS, fluorocitrate.

-A revision of Figure 6 is required (y axis)

Author Response

The manuscript entitled “Glial A2B adenosine receptors modulate abnormal tachykininergic responses and prevent enteric inflammation associated with high fat diet-induced obesity” by D’Antongiovanni and co-workers focuses on the role played by colonic A2BRs in the modulation of enteric glial cell functions. For this purpose they used a murine model of diet-induced obesity and they performed in vitro experiments mimicking obesity (rat transformed enteric glial cell treated with palmitate and LPS). The manuscript is well written, it is easy to comprehend and well organized.

However, I have some comments:

1. The authors used high fat diet model (8 weeks) to induce obesity in C57BL6/J mice. They should report body weight changes and food intake measure

As suggested by the Reviewer, food intake and body weight gain have been introduced into the revised version of the manuscript (see page 3 lines 111-112; page 7 lines 241-244; Figure 2 page 7 and page 8 lines 246-247).

2. The authors could add representative traces of contractile responses in Fig.2

As suggested by the Reviewer, representative traces of contractile responses have been introduced into Figure 3 of the revised manuscript (see page 9).

3. A quantification of fluorescent intensity is required about the effect of fluorocitrate on A2B receptor and glial protein S100β distribution assessed by confocal microscopy

As requested, the quantification of fluorescence intensity, expressed as density index of both A2B receptor and S100β, has been introduced into the revised manuscript (see page 5 lines 174-177; page 10 lines 274-275 and 277-278; Figure 4 page 11; page 12 lines 285-288).

4. The authors show that A2B receptor agonist, BAY60-6583, reduces TLR4 expression as well as IL-1β, GDNF and SP release induced by palmitate and lipopolysaccharide in cultured rat-transformed enteric glial cells. Exposure to LPS induces the expression of several inflammatory cytokines in enteric glia including IL-6 and TNF-α. Does the A2B agonist, BAY60-6583, modulate the levels of these cytokines in the in vitro model used?

We agree with the Reviewer that the exposure of enteric glial cells (EGCs) to LPS triggers the release of several cytokines, including IL-1β, IL-6 and TNF, as previously described (Rühl et al., Am J Physiol Gastrointest Liver Physiol, 2001; Murakami et al., J Neurosci Res, 2009), thus indicating that enteric glia plays a critical role in the immuno-inflammatory response of the enteric nervous system. However, in the present study we decided to focus the attention only on the release of IL-1β, since this cytokine, known to act as a first line of defense against invading microorganisms, is emerging as a critical player in several obesity-related comorbidities, such as insulin resistance, meta-inflammatory condition and cardiovascular diseases (Speaker et al., BMC physiology, 2012; Bing, Adipocyte, 2015). In parallel, it has been widely recognized that, under inflammatory conditions, IL-1β seems to play a pivotal role in sustaining the enteric tachykininergic transmission (Grider JR. Neurogastroenterol Motil, 2003).

At present, data about the effect of the A2BR agonist BAY60-6583 in counteracting the release of IL-6 and TNF from EGCs incubated with palmitate and LPS are lacking, and it represents clearly a point of interest, which deserves further investigations. Accordingly, we have already planned the performance of additional experiments, aimed at deepening the role of adenosine in shaping the interplay between enteric glia, enteric macrophages and IL-6. However, it is worth to note that a number of previous studies showed that the pharmacological stimulation of A2A or A2B receptors counteracted the release of pro-inflammatory cytokines (i.e. TNF, IL-6, IL-12) from human and murine macrophages (Haas, et al. Journal of cardiovascular translational research 2011; Hasko, et al. Journal of immunology 1996; Nemeth, et al.  Journal of immunology 2005; Hasko, et al. Experimental Biology 2000; Csoka, et al. Blood 2007). Based on this knowledge, it is conceivable that the stimulation of glial A2BRs could be able to modulate the release of IL-6 and TNF, also under our experimental conditions. In this regard, Csòka et al. (2014) reported that the pharmacological stimulation of A2BRs suppressed the pro-inflammatory response of macrophages induced by free fatty acids. In particular, the authors reported that the A2B receptor engagement counteracted the release of cytokines (i.e. IL-6 and TNF) and induced the polarization of such immune cells towards an anti-inflammatory phenotype (Csòka et al., FASEB journal, 2014).

As requested, we have introduced a sentence concerning the role of glial A2BRs in the modulation of IL-6 and TNF release under our experimental conditions into the “Discussion” section of the revised manuscript (see page 16 lines 437-438, 441 and page 17 lines 442-448).

5. The major signalling pathway of A2BAR is suggested to involve adenylyl cyclase. Could the authors explain the intracellular signalling pathways coupled to the A2B receptor activation in the effects observed? Does the selective targeting of TLR4 on enteric glial cells cause a downstream inhibition of nuclear factor kappa B-dependent inflammation?

This is an interesting point. It is widely recognized that A2BRs are Gs or Gq protein-coupled receptors (Antonioli et al., Pharmacology & therapeutics, 2008; Sheth et al. Int J Mol Sci, 2014). The A2BR activation, eliciting Gs protein activation, determines the activation of adenylyl cyclase, with a consequent increase in intracellular cAMP. By contrast, the A2BR engagement via Gq protein can activate phospholipase C, determining the activation of protein kinase C, followed by an increase in intracellular calcium concentration. Previous studies reported that the release of glial cell derived neurotrophic factor (GDNF) is finely regulated by a variety of signaling pathways, including also cAMP. Indeed, it has been observed that an increase in intracellular cAMP induces a reduction of GDNF release (Verity et al., Journal of Neurochemistry, 1998). In line with this evidence, we observed that the pharmacological activation of A2BRs induces a reduction of GDNF release from enteric glial cells (EGCs). With regard for substance P (SP), data about the molecular mechanisms underlying SP release from enteric glia are presently lacking. However, some studies have investigated SP release from neurons of the central nervous system (Rage et al., Pflugers Arch. 1987; Suvas, J Immunol. 2017). These studies showed that, once synthesized, SP is transported into large dense-core vesicles (LDCVs) and then released via exocytosis (Rage et al., 1987; Suvas, 2017). Exocytosis is regulated by several types of calcium channels (i.e. N-type and L-type calcium channels), whose activity can be modulated by A2BR activation (Rage et al., 1987; Takasusuki et al., Anesthesiology. 2011).

In the present study, we observed that A2BR stimulation reduced SP release from EGCs. In this context, it is recognized that the activation of A2BRs, increasing intracellular cAMP, shapes intracellular calcium release, thus regulating its intracellular levels (Moore et al., Am J Physiol. 1998). Therefore, it is conceivable that the modulatory action of A2BRs on SP release from EGCs depends on its regulatory role on intracellular calcium. However, future and more focused studies are needed to corroborate this hypothesis.

Our study showed also that, when stimulated pharmacologically, A2BRs counteracted the glial pro-inflammatory responses, reducing both TLR4 expression and IL-1β release from cultured EGCs. This evidence is in line with previous data from human and murine macrophages (Haas, et al. Journal of cardiovascular translational research 2011; Hasko, et al. Journal of immunology 1996; Nemeth, et al.  Journal of immunology 2005; Hasko, et al. Experimental Biology 2000; Csoka, et al. Blood 2007). Of note, TLR4 signaling triggers the activation of NF-kB, which controls the expression of several pro-inflammatory cytokine genes, including IL-1β (Kawai et al., Trends Mol Med. 2007; Liu et al., Signal Transduct Target Ther. 2017). Therefore, as previously observed in other cell lines (Huang et al., Eur Rev Med Pharmacol Sci. 2014), the inhibition of TLR4 on EGCs could inhibit the downstream signaling of NF-Ò¡B.

As requested, we have introduced the current hypotheses on signaling pathways into Figure 8 of the revised manuscript (see page 18).

6. The authors could add the concentration used for BAY60-6583 and MRS1754 in all figures or in their legend to help the reader as well as for PA, LPS, fluorocitrate.

As suggested by the Reviewer, the concentration used for palmitate, LPS, fluorocitrate and A2BR ligands has been introduced in the legend into the revised version of the manuscript.

7. A revision of Figure 6 is required (y axis)

As suggested by the Reviewer, the y axis of Figure 6 has been revised: the word “IL1β” has been modified into “IL-1β” (see the new Figure 7 page 14)

Reviewer 2 Report

see file attached

Author Response

D’Antongiovanni et al. studied the effect of A2B adenosine receptor agonist and antagonist on the contractile activity of colonic longitudinal smooth muscle preparations from mice fed with standard diet or high fat diet. It was found that the A2B antagonist MRS1754 enhanced electrically evoked tachykininergic contractions, whilst the agonist BAY60-6583 decreased the contractions of longitudinal smooth muscle preparations from mice fed with high fat diet, but not with standard diet. In enteric glial cells the authors found that A2B receptors modulate the release of interleukin-1β release, substance P and glial cell derived neurotrophic factor.

I have a few minor comments:

  1. The source of compounds, such as MRS1754, BAY60-6583, and fluorocitrate, needs to be provided.

As suggested, the source of fluorocitrate and A2BR ligands has been provided in the “Materials and Methods” section of the revised manuscript (see page 4 lines 137 and 141).

  1. The concentrations of the MRS1754, BAY60-6583, and some other compounds should be included in the figures or figure legends.

As suggested by the Reviewer, the concentrations used for palmitate, LPS, fluorocitrate and A2BR ligands have been introduced in the legend into the revised manuscript.

  1. Briefly mention the rationale that different concentrations of MRS1754 and BAY60-6583 were used in different experiments.

In the experiments on colonic preparations from standard diet (SD) and high fat diet (HFD) mice, the concentration of A2BR ligands were selected in accordance with a previous study conducted by Antonioli et al. (Purinergic signalling 2017). In this study, the authors performed a set of preliminary experiments to select the concentrations of BAY60-6583 and MRS1754 suitable for better appreciating the effects of test drugs on colonic contractile activity (Antonioli, L. et al., 2017). As suggested by the Reviewer, we have introduced a sentence concerning the concentrations used in functional experiments into the “2.1.3” section of the revised manuscript (see page 4 lines 145).

With regard for the experiments on cultured enteric glial cells, the concentrations of A2BR ligands were selected in accordance with a previous study conducted by Daniele et al. (Cell Death Dis 2014). In that study, the authors performed a dose-dependent study to select the concentration of BAY60-6583 and MRS1754 not affecting cell proliferation. On this basis, we decided to use the following concentrations of A2BR ligands in cell culture experiments: 0.05 μM BAY60-6583 and 0.1 μM MRS1754.

  1. Need statistical analysis to support the statement “In colonic preparations from SD mice, the incubation with A2BR ligands or FC did not alter electrically evoked NK1-mediated tachykininergic contractions (Figure 2A and 2B).” (page 7).

As requested, the statistical analysis to support the above mentioned statement has been introduced into the revised manuscript (see page 8 lines 261-262).

  1. In enteric glial cells, what is the A2B expression level in comparison with A1, A2A and A3 receptors?

A2BR is the adenosine receptor subtype mainly expressed both in human and rodent enteric glia, as compared with A1, A2A and A3 receptors (Christofi et al., The Journal of Comparative Neurology, 2001; Vieira et al., Neurochemistry International, 2011). In particular, it has been reported that A2BRs are highly expressed on enteric glial cells in human jejunum and colon, whereas A1 and A2A are absent or scarcely detectable (Christofi et al., 2001). In line with this data, it has been observed that in rodents A2BR is the subtype mainly expressed in myenteric glial cells, whereas A1, A2A or A3 are scarcely expressed (Vieira et al., 2011). In reply to the Reviewer’s query, a sentence about the A2BR expression on EGCs has been introduced into the revised manuscript (see page 2 lines 86-87).
